# The Role of Monocytes in the Natural History of Idiopathic Pulmonary Fibrosis: A Systematic Literature Review

**DOI:** 10.3390/ijms26136538

**Published:** 2025-07-07

**Authors:** Diego Lema, Esteban Kosak Lopez, Justin Lam, Irakli Tskhakaia, Yurilu Gonzalez Moret, Shahrzad Abdollahi

**Affiliations:** 1Department of Medicine, Jefferson Einstein Hospital, Philadelphia, PA 19141, USA; esteban.kosaklopez@jefferson.edu (E.K.L.); justin.lam@jefferson.edu (J.L.); irakli.tskhakaia@jefferson.edu (I.T.);; 2Division of Rheumatology, Jefferson Einstein Hospital, Philadelphia, PA 19141, USA

**Keywords:** idiopathic pulmonary fibrosis, monocytes, cytokines, insterstitial lung disease

## Abstract

Emerging evidence suggests a significant association between monocytes and the pathophysiology and prognosis of idiopathic pulmonary fibrosis (IPF). This review aims to systematically evaluate current knowledge regarding blood monocyte counts and their relationship with the etiology, progression, and prognosis of IPF. We conducted a systematic search in the PubMed database for articles published through 17 February 2025, using the MeSH terms “lung diseases, interstitial” and “monocytes,” which yielded 314 results. After filtering for full-text articles in English (n = 242), we included only studies focusing on blood monocyte counts with clinical implications in IPF. Articles relating to other cell types or non-IPF lung diseases were excluded. Our systematic search identified 12 relevant articles. Monocytes play an essential role in regulating inflammatory responses and resolution across multiple diseases, with established but incompletely understood contributions to lung fibrosis development in IPF. Correlations have been demonstrated between elevated blood monocyte counts and the following: (1) the presence and progression of interstitial lung abnormalities, (2) the progression from an indeterminate usual interstitial pneumonia (UIP) pattern on CT scans to definitive IPF, and (3) worse lung function parameters, an increased risk of acute exacerbations, and reduced overall survival in IPF patients. Monocytes serve as critical orchestrators throughout IPF’s natural history—from early interstitial changes to disease progression and acute exacerbations. Targeting monocyte recruitment pathways and reprogramming their differentiation represents a promising therapeutic approach, while circulating monocyte counts offer potential as accessible biomarkers for disease progression and treatment response. Future research should characterize stage-specific monocyte phenotypes to enable precision-targeted interventions.

## 1. Introduction

Idiopathic pulmonary fibrosis (IPF) is a progressive, devastating disease with poor outcomes, characterized by lung fibrosis in a specific radiographic and histopathologic pattern called usual interstitial pneumonia (UIP) [1]. Significant research efforts have focused on understanding the pathogenic mechanisms underlying IPF. The current prevailing hypothesis suggests that recurrent alveolar epithelial micro-injuries trigger abnormal repair processes in genetically susceptible older individuals [2,3].

Monocytes play a crucial role in normal lung injury repair processes [4]. However, monocyte-derived macrophages have been consistently implicated in pulmonary fibrosis development in murine models [5], suggesting a dual role for these cells in both tissue repair and fibrosis.

Monocytes are stably produced throughout life and perform diverse functions in the human body, including inflammation modulation, host defense, and tissue development. While primarily produced in the bone marrow, monocytes can also undergo extramedullary production during conditions such as psychosocial stress or malignancy. Additionally, the spleen serves as a reservoir of monocytes that can be rapidly deployed in response to acute inflammatory insults or tissue injuries [6].

Humans have at least three distinct monocyte subpopulations: classical (CD14^+^CD16^−^) monocytes comprising 80–90% of the circulating pool, nonclassical (CD14^lo^CD16^+^) monocytes, and intermediate (CD14^+^CD16^+^) monocytes constituting the remainder. In mice, Ly6C^hi^CX3CR1^int^CCR2^+^CD62L^+^CD43^lo^ monocytes correspond to human classical monocytes, while Ly6C^low^CX3CR1^hi^CCR2^Lo^CD62L^−^CD43^+^ monocytes are analogous to human nonclassical monocytes [7].

Classical monocytes predominantly exhibit pro-inflammatory functions, whereas nonclassical monocytes are more involved in tissue repair and patrolling behaviors. Intermediate monocytes are generally considered a transitional state between classical and nonclassical phenotypes [8].

Monocyte subsets are distinguished by the differential expression of surface markers and cytokine profiles. Classical monocytes highly express CCR2 (CD192) and produce elevated levels of granulocyte colony-stimulating factor (G-CSF), CCL2, and IL-10. Nonclassical monocytes express higher levels of CX3CR1 (fractalkine receptor) and secrete TNF-α and IL-1β [9].

Monocyte-derived macrophages differ from resident macrophages. The lung houses three main types of macrophages: alveolar macrophages located on the luminal surface of alveoli, interstitial macrophages situated within the lung parenchyma, and monocyte-derived macrophages that infiltrate during inflammation. Alveolar macrophages exhibit a unique transcriptional profile, whereas interstitial macrophages share transcriptional similarities with resident macrophages in other tissues [10]. The third type, derived from classical monocytes, emerge during injury responses, either supplementing or replacing resident macrophages [11].

Evidence indicates that alveolar macrophages are derived from fetal progenitors and self-maintain throughout life [12]. While interstitial macrophages may share similar developmental origins, this remains incompletely characterized [13]. Resident macrophages serve as essential regulators of lung microenvironmental homeostasis, whereas monocyte-derived macrophages of myeloid origin primarily orchestrate inflammatory responses following tissue injury.

While the precise triggers for IPF development remain elusive, monocytes have emerged as key players in both health and disease across multiple tissue types [14]. A growing body of evidence demonstrates the active participation of monocytes in lung fibrosis pathogenesis [15]. Recent clinical reports have consistently noted associations between elevated blood monocyte counts and clinical outcomes in IPF. This review aims to cover the current knowledge regarding monocytes’ roles throughout the natural history of IPF, from early disease to progression and exacerbation.

## 2. Methods

We conducted a systematic literature search in the PubMed database for articles published through 17 February 2025, using the MeSH terms “lung diseases, interstitial” and “monocytes,” which yielded 314 initial results. We applied the following inclusion criteria: (1) full-text articles available in English, (2) studies focused on monocytes in idiopathic pulmonary fibrosis, and (3) research examining associations between blood monocyte counts and clinical implications in IPF. We excluded studies focused on other cell types or investigating monocytes in non-IPF interstitial lung diseases. After applying filters for full-text availability and English language (n = 242), we further screened articles based on their title and abstract review, followed by a full-text assessment to identify the final set of relevant publications. Figure 1 summarizes this process.

This systematic review of the literature was performed according to the PRISMA guidelines and PRISMA checklist. The PRISMA flow diagram can be found below. (Figure 2). This systematic review was registered to the pen science framework (https://osf.io/).

## 3. Results

### Studies and Clinical Trials Associate Monocytes with ILD Diagnosis, Progression and Activity

Our systematic search identified 12 articles that provided significant contributions to understanding the relationship between monocytes and idiopathic pulmonary fibrosis.

The potential role of monocytes in lung fibrosis pathogenesis has been the subject of prior studies, with early studies aiming to better define the etiological factors underlying fibrotic lung diseases [16].

Prospective studies involving apparently healthy individuals without known interstitial lung disease (ILD) have been performed to evaluate important associations between monocyte counts and early lung abnormalities. Kim et al. [17] demonstrated that elevated blood monocyte counts correlated with the presence of interstitial lung abnormalities (ILA) across four independent longitudinal cohorts comprising 7396 subjects. Notably, higher monocyte counts also predicted ILA progression over a 5-year period in one cohort (n = 3547) and were associated with reduced forced vital capacity (FVC) in two cohorts (n = 3203) [17].

In a significant retrospective analysis, Achaiah et al. [18] reported that approximately half of their patient cohort with initially indeterminate UIP patterns on chest CT scans progressed to develop definitive IPF over a mean follow-up period of 4 years. This progression was strongly associated with elevated blood monocyte levels measured at the time of initial CT imaging [18].

IPF patients demonstrate significantly higher blood monocyte counts compared to healthy controls [19]. Within IPF cohorts, patients with elevated monocyte counts exhibit worse diffusing capacity for carbon monoxide (DLCO) measurements and poorer overall survival [20,21]. Furthermore, an elevated monocyte count has been found to be a biomarker for disease progression, hospitalization risk, and mortality in IPF patients [19,22].

Bernardinello et al. [23] confirmed the negative correlation between baseline monocyte counts and both FVC and disease progression and introduced the lymphocyte-to-monocyte ratio (LMR) as a prognostic indicator. Using a threshold of 4.18, they demonstrated that newly diagnosed IPF patients with an LMR below this cutoff experienced significantly worse survival [23]. Interestingly, the LMR was particularly reduced in patients with comorbid IPF and lung cancer compared with those with either newly diagnosed or end-stage IPF alone [23]. In contrast, Barratt et al. found that LMR did not predict survival in fibrotic hypersensitivity pneumonitis [24], potentially highlighting a specific pathophysiological role for monocytes in IPF.

Elevated blood monocyte counts also demonstrate a predictive value for acute exacerbations in IPF patients. Kawamura et al. observed that patients with increased monocyte counts at antifibrotic medication initiation developed acute exacerbations more frequently and within shorter timeframes compared with those with normal monocyte levels [25]. Critically, multivariate analysis confirmed an elevated monocyte count as an independent risk factor for acute exacerbations even after adjustment for established predictors such as ILD-GAP score and FVC [25].

Fernandez and Kass highlighted a critical limitation in the current research: standard complete blood count measurements provide total monocyte numbers without distinguishing between functionally distinct monocyte subpopulations [26]. They emphasized the importance of characterizing specific monocyte subsets to better elucidate which populations are most directly involved in IPF pathophysiology [26].

Advancing this concept of monocyte subset characterization, Yamashita et al. [27] employed flow cytometry to examine classical monocyte phenotypes as potential biomarkers for differentiating IPF from nonspecific interstitial pneumonia (NSIP). They observed that S100A9^+^CD163^−^ classical monocytes were significantly reduced in NSIP compared to IPF, while S100A9^−^CD163^+^ monocytes were markedly elevated in NSIP relative to both IPF patients and healthy controls [27]. Within the IPF cohort, they noted a trend toward the correlation between S100A9^+^CD163^−^ monocyte percentages and serum surfactant protein-D (SP-D) levels, an established biomarker of alveolar damage [27]. Multivariate regression analysis confirmed that the proportions of both S100A9^+^CD163^−^ and S100A9^−^CD163^+^ cells were independently associated with IPF diagnosis [27]. These findings suggest that distinct monocyte phenotypic profiles may exist even among different fibrotic idiopathic interstitial pneumonias, potentially reflecting divergent pathogenic mechanisms.

Chauvin et al. considered studying soluble CD163 as a potential prognostic biomarker, given its high expression in monocyte-derived macrophages and its established role as a marker of macrophage activation [28]. Their analysis failed to demonstrate any significant association between serum CD163 levels and the prognosis in IPF patients [28].

Table 1 summarizes key studies that have found an association between monocytes and IPF development and progression.

## 4. Discussion

### 4.1. Immunophenotypes and Cytokines of IPF Monocytes

The basic phenotypic classification of human monocytes encompasses three distinct subsets: classical (CD14^+^CD16^−^), non-classical (CD14^dim^CD16^+^), and intermediate (CD14^+^CD16^+^) monocytes [29]. In murine models, classical monocytes are characterized as Ly-6C^+^CCR2^hi^, while non-classical monocytes exhibit a Ly-6C-CCR2^lo^ phenotype. Functionally, these subsets demonstrate specialized capabilities: classical monocytes excel in phagocytosis and robust reactive oxygen species (ROS) production; non-classical monocytes, with their high expression of chemokine receptors, patrol the vasculature and infiltrate inflamed tissues where they secrete various inflammatory mediators including IL-6, IL-8, CCL2, CCL3, and CCL5; and intermediate monocytes demonstrate enhanced antigen presentation capacity [29]. Within specific tissue microenvironments and disease processes, including IPF, infiltrating monocytes exhibit subtle but significant transcriptional and phenotypic adaptations. Upon tissue infiltration, monocytes may differentiate into macrophages with diverse phenotypes. While the dichotomous M1/M2 polarization paradigm represents an oversimplification, macrophages generally exist along a spectrum between pro-inflammatory M1 phenotypes (characterized by phagocytic activity) and M2 phenotypes involved in both inflammation resolution and fibrosis promotion [30].

Murine models have provided crucial insights into monocytes’ pivotal role in pulmonary fibrosis pathogenesis. Following lung injury, levels of Ly-6C^hi^ circulating monocytes increase reaching a plateau phase. The inflammatory cascade orchestrated by these infiltrating monocytes ultimately promotes a shift into an alternative M2-like alveolar and interstitial activation state that favors fibrotic processes [31].

These observations align with clinical findings in pulmonary fibrosis patients, who demonstrate predominant M2-polarized alveolar macrophage populations driven by cytokines such as IL-4 and IL-10 [32]. Multiple independent murine models have corroborated this M2-skewing phenomenon, further supporting its pathophysiological relevance [33,34].

Several investigators have characterized monocyte phenotypes in IPF patients. Fraser et al. [35] employed multi-color flow cytometry on lung specimens and identified an enhanced expression of CD64 (FCγR1) on monocytes—particularly classical monocytes—which correlated with fibrosis severity and augmented responsiveness to type I interferons ex vivo. Their analysis of IPF patients’ serum revealed elevated levels of CCL-2, CSF-1, and IL-6. Complementary single-cell transcriptomic analyses of IPF lungs demonstrated the enrichment of CD64^hi^ monocytes and “transitional macrophages” with an upregulated expression of CCL-2 and type I interferon-responsive genes [35].

Yamashita et al. reported that classical monocytes from IPF patients expressed higher levels of S100A9 and reduced CD163 compared to controls. They hypothesized that this phenotypic profile enhances reactive oxygen species production, potentially perpetuating non-resolving tissue damage and promoting long-term fibrotic remodeling [27].

The unique features of macrophages during inflammatory exacerbations have been studied and described, including distinctive lipid metabolism profiles characterized by apolipoprotein E (ApoE) expression—which facilitates apoptotic cell clearance—and the upregulation of pro-fibrogenic genes including fibronectin, osteopontin, and TGF-β1 [36]. Mechanistically, monocyte-derived macrophages interact with lipid ligands via triggering receptors expressed on myeloid cells (TREM), inducing anti-apoptotic and pro-fibrotic responses [37]. Both circulating monocytes and tissue-resident alveolar macrophages (TR-AMs) upregulate the scavenger receptor MARCO, which works synergistically with ApoE to enhance their phagocytic capacity [38].

Type I interferon’s signaling involvement in IPF pathogenesis has been corroborated by multiple investigators. The IL-1R-associated kinase M (IRAK-M) has been shown to regulate pulmonary fibrosis development in a bleomycin-induced pulmonary fibrosis (BLM-IPF) murine model [39]. They found that IRAK-M-deficient mice exhibited significantly reduced monocyte-derived macrophage infiltration into the lungs following a bleomycin challenge. Mechanistically, this reduction resulted from impaired CCR2 upregulation on monocytes in the absence of IRAK-M, consequently attenuating lung fibrosis development [39]. These findings establish IRAK-M as a critical molecular link between inflammatory signaling, monocyte recruitment, and fibrotic progression.

### 4.2. Monocyte Migration to IPF Lungs and Differentiation into Pro-Fibrotic Macrophages

The CCL2–CCR2 signaling axis plays a central role in monocyte cell migration. CCL2 binds to CCR2 on monocyte surfaces, with CCR2^+^ monocytes exhibiting highly pro-inflammatory properties. In murine models, CCL2 serves as a principal chemokine driving classical monocyte migration to the lung, and as previously noted, IPF patients demonstrate significantly elevated serum CCL2 levels [35]. Recent evidence indicates that bronchoalveolar lavage CCL2 concentrations correlate with radiographic fibrosis severity in post-COVID-19 pulmonary fibrosis [40], suggesting a conserved mechanism across different fibrotic lung pathologies. Following tissue infiltration, monocytes can undergo local proliferation through colony-stimulating factor 1 receptor (CSF1R)-dependent mechanisms, subsequently differentiating into interstitial macrophages under the regulation of the MafB transcription factor [41]. Critical to their pro-fibrotic capacity, Notch2 signaling directs their differentiation into a specialized macrophage lineage implicated in pulmonary fibrosis [42].

Osterholzer et al. [43] elucidated a CCR2-dependent pathway through which classical (Ly-6C^hi^) monocytes may induce pulmonary fibrosis following alveolar type 2 epithelial cell injury [42]. In their transgenic mouse model selectively depleted of type 2 alveolar cells, the researchers observed significant inflammation mediated by CD11b^+^ non-resident macrophages and their classical monocyte precursors [43]. Notably, CCR2-deficient mice demonstrated protection against fibrosis development, weight loss, and mortality [43]. Mechanistic analyses revealed that these infiltrating non-resident macrophages and monocytes expressed pro-fibrotic mediators including IL-13 and TGF-β, upregulated collagen genes, and contained intracellular collagen [43]. In a parallel clinical observation, Puukila et al. demonstrated in a cross-sectional study that CCL2-driven classical monocyte migration promoted M2 macrophage accumulation in the lungs of chronic heart failure patients, potentially contributing to pulmonary fibrotic remodeling and consequently exacerbating dyspnea [44].

While CCR2^+^ monocytes are predominantly recognized for their pro-inflammatory functions, mounting evidence indicates they may also exert immunosuppressive effects in certain contexts. Lebrun et al. characterized a distinct population of CCR2^+^ myeloid-derived suppressor cells in a silica-induced pulmonary fibrosis model that suppressed T lymphocyte proliferation in vitro [45]. These cells infiltrated the lung parenchyma prior to fibrosis onset, retained their monocytic phenotype without further differentiation, and secreted substantial amounts of TGF-β [45]. This TGF-β production subsequently enhanced the fibroblast expression of the tissue inhibitor of metalloproteinase-1 (TIMP-1), which inhibits matrix metalloproteinase collagenolytic activity, thereby promoting collagen accumulation and fibrosis progression [45].

Beyond the CCL2–CCR2 axis, CX3CL1 (fractalkine) provides another critical molecular signal orchestrating monocyte recruitment to fibrotic lungs. This chemokine interacts with CX3CR1, which is highly expressed on non-classical monocytes (CD14^+^CD16^++^ and CCR2^lo^CX3CR1^hi^) [45]. In fibrosing interstitial lung diseases, CX3CL1 establishes concentration gradients within the pulmonary microenvironment that drive the recruitment of these non-classical monocyte populations [46,47]. Figure 1 summarizes the key steps in monocyte recruitment to the lung, proliferation, differentiation, and eventually IPF development and progression (Figure 1).

Recent single-cell analyses have uncovered additional intercellular communication networks mediating monocyte recruitment in IPF. Fibroblasts secrete CXCL12 (stromal cell-derived factor 1), which acts on CXCR4^+^ monocytes to activate extracellular signal-regulated kinase (ERK) signaling pathways, driving their recruitment into the lung [48]. Notably, the abundance of these CXCR4^+^ infiltrating monocytes was significantly higher in IPF patients with reduced forced vital capacity (FVC), suggesting a potential relationship with disease severity [48].

### 4.3. Monocytes Induce Local Pulmonary Fibrosis

Monocytes serve dual roles in pulmonary fibrosis pathophysiology, functioning both as inflammatory mediators and direct contributors to fibrotic remodeling. Their accumulation within fibrotic lung tissue follows initial epithelial injury and subsequent chemoattractant gradient formation, culminating in their recruitment and differentiation into monocyte-derived alveolar macrophages (Mo-AMs) [49,50]. This recruitment process depends predominantly on chemokine signaling networks, particularly CCL2-CCR2 interactions, which facilitate monocyte differentiation into Mo-AMs that progressively become the predominant macrophage population within fibrotic lungs [51].

Importantly, these Mo-AMs exhibit distinct transcriptional profiles from TR-AMs, characterized by the enhanced expression of pro-fibrotic genes and inflammatory mediators [52]. Unlike Mo-AMs, TR-AMs represent self-renewing populations that maintain their numbers independently of peripheral monocyte influx [53].

A strong association between peripheral monocytosis and poor prognoses in IPF patients has been described. Elevated circulating monocyte counts, increased expressions of monocyte-specific genes, reduced T-cell subpopulations, and attenuated T-cell costimulatory pathways collectively predict an increased mortality risk and worse clinical outcomes [54]. Conversely, experimental depletion of circulating monocytes in preclinical models significantly attenuates fibrosis development, highlighting their causal role in disease pathogenesis [55].

The coexistence of functionally distinct macrophage populations within the fibrotic lung microenvironment drives complex polarization dynamics. Under persistent inflammatory stimuli, infiltrating monocytes differentiate into Mo-AMs that adopt predominantly pro-fibrotic phenotypes characterized by the enhanced secretion of transforming growth factor-β (TGF-β) and direct collagen production [56]. This polarization process creates a critical imbalance between pro-inflammatory and anti-inflammatory/pro-resolving macrophage subsets, perpetuating fibrotic remodeling. Clinical investigations consistently demonstrate that elevated monocyte counts and increased monocyte-specific gene expression signatures correlate with higher mortality rates and disease severity [53], while experimental therapeutic approaches targeting circulating monocytes have shown promising anti-fibrotic effects [54].

Beyond direct pro-fibrotic functions, monocytes and Mo-AMs orchestrate complex crosstalks with fibroblasts through cytokine networks. While normal human lung fibroblasts modulate monocyte cytokine responses to maintain homeostasis, fibrotic fibroblasts display dysregulated tumor necrosis factor-α (TNF-α) receptor expression patterns and aberrant prostaglandin E_2_ (PGE_2_) production [57,58]. These alterations further propagate chronic inflammatory signaling and fibrotic cascades. Mechanistically, granulocyte-macrophage colony-stimulating factor (GM-CSF) serves as a critical regulator of monocyte-driven fibrosis through prostaglandin-dependent pathways [59].

Recent technological advancements in spatial transcriptomics have begun to elucidate the complex topographical organization of monocyte-derived cells within fibrotic lung tissue. Studies demonstrate that distinct macrophage populations, many derived from peripheral monocytes, establish precise spatial niches within fibrotic lungs where they engage in specialized interactions with fibroblasts, epithelial cells, and other immune cell populations [60]. At the molecular level, the zinc finger protein ZC3H4 has emerged as an important monocyte regulator capable of attenuating pulmonary fibrosis progression through interleukin-10 (IL-10)-mediated anti-inflammatory mechanisms [61], highlighting potential molecular targets for therapeutic intervention.

Intriguingly, monocytes demonstrate duality in their roles in fibrosis, potentially contributing to both fibrosis progression and resolution depending on microenvironmental cues. Mesenchymal stromal cell-derived exosomes and tissue-resident alveolar macrophages can reprogram monocyte phenotypes toward anti-fibrotic profiles characterized by reduced TGF-β expression and enhanced IL-10 production, effectively reversing established fibrosis in experimental models [62]. Mechanistically, lung Mo-AMs express ApoE, which facilitates the efficient clearance of apoptotic cells and promotes fibrosis resolution [63,64].

Familial clustering studies reveal that first-degree relatives of patients with fibrotic idiopathic interstitial pneumonias (IIPs), even without previously diagnosed ILD, have, approximately, a 1-in-6 chance of harboring undiagnosed ILD and frequently demonstrate ILA [65]. Within a carefully characterized cohort of first-degree relatives of fibrotic IIP patients, those with ILA were predominantly male and exhibited significantly higher absolute monocyte counts at baseline, accompanied by reduced FEV_1_/FVC ratios and decreased percent-predicted measurements of FVC, TLC, and DLCO [66].

These observations have led to the development of blood-based biomarker panels validated across multiple independent cohorts. Proteomic analyses have established a multivariable predictive model incorporating growth differentiation factor 15 (GDF15), surfactant proteins D (SFTPD) and B (SFTPB), WAP four-disulfide core domain protein 2 (WFDC2), CUB domain-containing protein 1 (CDCP1), and stratifin (SFN) [66]. This panel can detect ILA in at-risk relatives with high accuracy, potentially reducing reliance on chest CT scanning [67].

Mechanistically, monocytes contribute to GDF15 elevation in IPF through their differentiation into macrophages that both produce and respond to GDF15, creating a feed-forward loop that promotes fibrosis via fibroblast activation and further macrophage recruitment and polarization [68,69,70]. These findings collectively indicate that monocyte-related processes actively participate in the early pathophysiological events leading to pulmonary fibrosis development.

Overall, monocytes serve as key regulators of pulmonary fibrosis, with their roles varying based on local microenvironmental signals. By targeting monocyte recruitment, differentiation, and cytokine signaling pathways, novel therapeutic approaches may be developed to mitigate fibrosis and improve patient outcomes.

### 4.4. Progression from Pre-IPF to IPF: Role of the Monocyte

ILA are increasingly recognized as potential precursor lesions in IPF development [71]. The established correlation between elevated blood monocyte counts and both the presence and progression of these early interstitial changes [17] raises fundamental questions about the temporal evolution of disease pathophysiology and suggests potential opportunities for early therapeutic intervention before irreversible fibrosis develops.

An indeterminate UIP pattern on chest CT—another potential pre-IPF radiographic phenotype—has also demonstrated associations with elevated blood monocyte counts. Longitudinal studies reveal that progression from an indeterminate UIP to a definitive UIP-pattern IPF over a mean follow-up period of 4 years correlates significantly with sustained elevations in circulating monocytes [22]. Given IPF’s heterogeneous natural history, ranging from a rapidly progressive disease culminating in respiratory failure within months to indolent courses spanning many years [1], persistent monocytosis may reflect an ongoing pathophysiological process continually recruiting monocytes to the lung microenvironment, thereby perpetuating and accelerating fibrotic remodeling.

Multiple studies have consistently identified elevated blood monocyte counts as a predictor of worse lung function parameters (including DLCO and FVC), reduced survival rates, and an increased risk of acute exacerbations in established IPF patients [20,22]. Classical monocytes specifically are expanded in both stable and progressive IPF patients compared to matched controls, with the most pronounced elevations observed in rapidly progressive disease [72]. These observations suggest a potential mechanistic role for classical monocytes in driving disease progression. Additionally, transcriptomic analyses have identified evidence for an active lung-blood immune recruitment axis involving CCL7, a potent monocyte chemoattractant [72]. Collectively, these findings underscore the importance of monocyte dynamics in both IPF pathogenesis and progression.

The Interstitial Lung Disease-Gender, Age, and Physiology (ILD-GAP) score represents a validated clinical prediction model for mortality risk stratification across various ILD subtypes. This model extends the original GAP index developed specifically for IPF by incorporating four key variables: gender (G), age (A), and two physiological parameters (P)—forced vital capacity (FVC) and diffusing capacity for carbon monoxide (DLCO) [73]. Recently, Hirata et al. developed an enhanced prognostic model incorporating blood monocyte ratio alongside traditional GAP parameters [74]. Their analysis demonstrated that this monocyte-inclusive model provided superior prognostic accuracy compared with the standard ILD–GAP index [74], further substantiating the clinical utility of circulating monocytes as biomarkers reflecting underlying pathobiological processes within the fibrotic lung.

In the perioperative context, patients undergoing a surgical lung biopsy for ILD diagnosis exhibit noteworthy monocyte dynamics. Postoperative elevations in monocyte counts demonstrate strong correlations with prolonged hospitalization and reduced FVC at 3-month follow-up assessments [75]. Within this cohort, day 1 postoperative monocyte counts were significantly elevated compared to preoperative, perioperative, and day 90 postoperative values [75]. Furthermore, postoperative monocyte count elevations were substantially more pronounced in patients undergoing open thoracotomy procedures compared to those receiving less invasive video-assisted thoracoscopic surgery (VATS) [75], suggesting that the magnitude of surgical trauma may influence monocyte mobilization and potentially impact subsequent clinical outcomes.

Elevated blood monocyte counts have also demonstrated associations with antifibrotic treatment outcomes and adverse event profiles. Tsuneyoshi et al. reported that IPF patients with high baseline monocyte counts required particularly vigilant monitoring for nintedanib-associated adverse effects [76]. Moreover, elevated monocyte count emerged as an independent risk factor for nintedanib treatment failure [76]. Interestingly, their analysis found no significant differences in FVC decline trajectories or acute exacerbation frequencies across different nintedanib dosing regimens [76], suggesting that monocyte-related mechanisms might influence treatment responses independently of conventional dosing considerations.

Lower baseline blood monocyte counts, particularly when combined with younger age, predict superior functional improvement in fibrotic ILD patients [76]. Interestingly, in patients with mixed inflammatory and fibrotic disease components receiving anti-inflammatory therapies, age emerged as the strongest predictor of response [77]. These observations collectively suggest that blood monocytes serve as active participants throughout the entire natural history of fibrotic lung disease—from early interstitial changes through established fibrosis and eventual progression—with their relative pathophysiological contributions potentially varying based on the predominant disease phenotype.

This converging evidence compels reconsideration of fibrotic lung disease as a continuum, beginning far earlier than clinically apparent disease (Figure 1). Monocytes appear to contribute to pathogenesis from the earliest detectable interstitial changes through progressive fibrotic remodeling, IPF diagnosis, disease progression, and ultimately to decreased survival [78]. Understanding these monocyte-driven mechanisms across the disease spectrum creates opportunities for novel therapeutic approaches targeting specific pathways at different disease stages.

### 4.5. Potential Therapeutic Approaches

Monocytes and their differentiated macrophage derivatives have emerged as promising therapeutic targets in ILD and IPF. While traditional research on approved antifibrotic medications like nintedanib initially focused their direct effects on fibroblast activity, recent investigations have uncovered significant impacts on monocyte–macrophage biology. Nintedanib demonstrates multifaceted effects on monocytes, including reduced CCR2^+^ cell infiltration into fibrotic lungs [79] and the inhibition of M2 macrophage differentiation [80,81], ultimately attenuating pro-fibrotic activation cascades. Similarly, pirfenidone—the other approved antifibrotic agent—indirectly suppresses fibroblast activity through the inhibition of M2 macrophage activation [82]. Mechanistically, pirfenidone also downregulates COL1A1 expression in circulating monocytes [83], further highlighting the importance of monocyte-targeted effects in current therapeutic approaches. In a related note, pirfenidone downregulates nuclear factor kappa-B (NF-κB) p50 in macrophages, reducing M2 polarization and in doing so, ameliorating radiation-induced lung fibrosis [84].

Mycophenolate mofetil, commonly employed in the management of connective tissue disease-associated ILD (CTD-ILD), exhibits significant effects on macrophage biology, reducing both infiltration and the viability of these cells [85]. The modulatory activities of mycophenolate and other immunosuppressive agents on the monocyte–macrophage axis likely contribute substantially to their therapeutic efficacy in inflammatory ILDs [86,87], suggesting that targeted enhancement of these mechanisms could potentially improve clinical outcomes.

Other more experimental agents are being developed to interfere with monocyte and macrophage contributions to pulmonary fibrosis and thus attenuate pathology. While yet in preclinical stages, they warrant attention as they show promise and could be developed for clinical studies. Many of these agents interfere with macrophage infiltration and their M2 polarization, which are drivers of IPF progression. Neotuberostemonine, an alkaloid that reduces macrophage recruitment to the lungs and inhibits their M2 polarization, decreases BLM-IPF [88]. Icariside II similarly halts M2 polarization and alleviates BLM-IPF, but does so by modulating the PI3K/Akt/β-catenin signaling pathway [89]. Disulfiram is being investigated as it has a similar effect on M2 macrophages via FROUNT inhibition, which regulates CCR2 signaling [90]. Finally Schisandra, an herbal compound, has been found to also reduce M2 polarization and protect against BLM-IPF [91]. Another druggable target in monocytes is SHP-1. A drug that inhibited this target was found to suppress CSF1R signaling in M2 macrophages in the lung, which improved BLM-IPF [92].

Strategies aimed at blocking monocyte migration or differentiation have shown promise in preclinical studies. The inhibition of key molecules in this pathway, such as S100A4 (critical for M2 activation) and β-catenin (which activates TGF-β), can dampen fibroblast activation and thus reduce collagen deposition and fibrosis [93,94,95]. In addition, IL-10 inhibition using methyl palmitate has shown potential in decreasing fibrosis in vitro [96,97]. These findings suggest that therapies directed at monocyte–macrophage pathways may slow disease progression.

Several ongoing clinical trials are evaluating antifibrotic approaches but do not specifically target the monocyte–macrophage axis [98,99,100]. Concurrently, rapid advances in gene editing technologies, particularly CRISPR-Cas9 systems, offer unprecedented opportunities to correct or silence aberrant genes involved in pathological monocyte functions [101,102]. The therapeutic landscape is further expanding through a combination of approaches integrating monocyte-targeted strategies with established antifibrotic agents, immunomodulators, or regenerative medicine interventions such as mesenchymal stem cell transplantation [103,104]. As our understanding of monocyte biology in fibrotic lung disease continues to evolve, these targeted interventions may be increasingly integrated into personalized treatment regimens, potentially transforming management paradigms for patients with ILD and pulmonary fibrosis.

One of the most cutting-edge therapeutic approaches being investigated in IPF involves phenotypic reprogramming utilizing small particles. Chang et al. [105] loaded monocyte-derived multipotent cells with nanoparticles carrying astaxanthin and trametinib. These monocyte-derived cells home to the lungs naturally, and their cargo was able to inhibit lung myofibroblast activation, repair damaged alveolar epithelial cells type II, and reverse IPF in mice [105]. Mansouri et al. were able to prevent or even reverse BLM-IPF by reprogramming pathogenic lung monocyte populations using mesenchymal cell-derived exosomes [62].

### 4.6. Role of Monocytes in Connective Tissue Disease ILD

While research examining monocyte contributions to idiopathic pulmonary fibrosis has expanded substantially, investigations into their role in CTD-ILD remain comparatively limited. Nevertheless, recent studies have begun to elucidate important monocyte-related mechanisms in these complex disorders.

A particularly compelling area of emerging research is the potential role of monocytes and other subtypes of inflammatory cells in the development of CTD-ILD. Several studies have explored this connection.

Bernstein et al. conducted a detailed analysis of the relationship between baseline absolute monocyte count (AMC) and forced vital capacity (FVC) decline in 136 systemic sclerosis-related ILD (SSc-ILD) patients enrolled in the phase 3 focuSSced trial [106]. Their results demonstrated a statistically significant inverse association between baseline AMC and FVC changes at 48 weeks, suggesting that AMC could serve as a clinically relevant biomarker for disease progression prediction in SSc-ILD [106]. This association appeared particularly robust in patients with early SSc characterized by elevated circulating inflammatory markers [106], potentially identifying a subgroup who might benefit most from targeted therapeutic interventions.

Mathai et al. [107] characterized the profibrotic phenotype of circulating monocytes in SSc-ILD patients compared to healthy controls [99]. Using comprehensive flow cytometric analyses, they quantified circulating collagen-producing cells and demonstrated the significant expansion of these populations in SSc-ILD patients [107]. Concurrently, plasma profiling revealed elevated levels of key inflammatory mediators including IL-10, MCP-1, IL-1RA, and TNF [107]. These findings collectively highlight the increased presence of fibrocytes, profibrotic monocytes, and fibrosis-associated mediators in the peripheral blood of SSc-ILD patients, providing novel mechanistic insights into disease pathophysiology and potential therapeutic targets.

Soldano et al. investigated nintedanib’s effects on profibrotic M2 phenotypes in cultured monocyte-derived macrophages isolated from SSc-ILD patients [81]. Their findings demonstrated that nintedanib treatment significantly reduced both gene expression and protein synthesis of key M2 macrophage surface markers [81]. Importantly, nintedanib downregulated critical profibrotic mediators including TGF-β1 and Mer tyrosine kinase (MerTK)—a receptor implicated in pulmonary fibrosis pathogenesis [81]. These observations suggest that nintedanib’s therapeutic effects in SSc-ILD may be substantially derived from the modulation of monocyte–macrophage phenotypes in addition to its established anti-fibroblast activities.

Examining a different CTD-ILD subtype, Poole et al. characterized inflammatory cell populations and myeloid-derived suppressor cells (MDSCs) in rheumatoid arthritis-related ILD (RA-ILD) [108]. Their analyses revealed the significant expansion of multiple myeloid cell subpopulations in RA-ILD patients, including CD16^+^ monocytes, monocytic MDSCs, and neutrophils [108]. Additionally, they observed increased eosinophil populations, suggesting potential contributions to disease pathogenesis and identifying novel therapeutic targets [108]. In a complementary investigation, this research group demonstrated an upregulated expression of genes involved in inflammation, fibrosis, epigenetic modification, and macrophage activation within circulating monocytes isolated from RA-ILD patients [109], further highlighting the central role of monocyte dysregulation across diverse CTD-ILD subtypes.

Clinical trials evaluating therapeutic agents in CT-ILD is a field in its infancy, with the studies available being scarce, yet promising. Mycophenolate mofetil (MMF) shows anti-fibrotic benefits across diverse settings. In an MDA-5 amyopathic dermatomyositis–ILD patient, adding MMF (1.5 g/day) improved PF ratio and lowered FGF-2, CX3CL1, IL-1ra, IL-17A, IP-10 and MCP-1 [110]. In NZB/W F1 lupus-nephritis mice, MMF plus rapamycin improved nephritis, suppressed mTOR/ERK signaling, and reduced TGF-β1, MCP-1, α-SMA, fibronectin and collagen [111]. In systemic sclerosis, mycophenolic acid induces monocyte/macrophage apoptosis, blunts IL-4–driven activation, and diminishes dermal myeloid signatures, inhibiting fibroblast activation [85]. These mechanistic insights align with and help explain the growing body of clinical evidence supporting MMF’s therapeutic efficacy in fibrotic lung disease.

A retrospective series in CT-ILD has shown that MMF significantly stabilizes or improves FVC and DLCO over 2–3 years, with a favorable tolerability profile [86,112]. Moreover, open-label trials such as LOTUSS confirm pirfenidone’s safety in SSc ILD, even with concurrent MMF [113]. The Scleroderma Lung Study III pilot (MMF + pirfenidone vs. MMF alone) reported a more rapid FVC improvement in the combination arm within 6 months, alongside trends toward better HRCT fibrosis scores and patient-reported outcomes [114]. These findings suggest that early combination antifibrotic and immunomodulatory regimens could more effectively slow disease progression, offering practical, evidence-based strategies.

Currently, there are no active clinical trials specifically investigating agents that directly target the monocyte–macrophage pathway in CTD-ILD. However, other investigational therapies such as Janus Kinase inhibitors (tofacitinib) and Interleukin-6 inhibitors (tocilizumab) may exert indirect effects on monocyte activity and are being explored in related fibrotic and inflammatory conditions [115,116,117]. Despite these developments, current clinical practice continues to follow established recommendations: the 2023 ACR/Chest guidelines endorse mycophenolate mofetil as the preferred first-line agent for the treatment of CTD- ILD [118].

## 5. Conclusions

Several emerging evidence links circulatory changes in monocyte counts and their phenotypic characteristics with the development and progression of IPF. This is perhaps not surprising, given the well-established role of monocytes as one of the first responders to tissue damage and their ability to communicate with other cellular populations, such as fibroblasts, to induce a cycle of non-resolving inflammation and tissue repair that results in fibrosis. Multiple authors have elucidated critical contributions of monocytes to IPF in both murine models and human subjects, from chemotaxis, proliferation, differentiation, and fibrosis. A deeper understanding of the basic immunology of monocytes in IPF is sure to continue to yield targetable steps and players in the immunopathogenesis of IPF that may result in promising therapeutic agents. Such efforts are already underway and are critical to halt the progression of such a debilitating disease as IPF. Most IPF therapies aim to reduce fibrosis by acting on fibroblast growth factors, but it is clear that monocyte contributions to this disease leave a wealth of targetable molecules that, as we have reviewed, play a role in IPF. The CCR2/CCL2 axis seems to be a particularly central chemokine-receptor pathway necessary for culprit monocyte infiltration into the lungs prior to disease development and could represent a promising treatment strategy for this devastating disease.

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
