# Peer review of "The Role of Monocytes in the Natural History of Idiopathic Pulmonary Fibrosis: A Systematic Literature Review"

_ijms, 2025, doi:10.3390/ijms26136538_

Round 1

Reviewer 1 Report

Comments and Suggestions for Authors

I  would like to see an expansion on the section about therapeutics that could be used as a result of knowing about the role of monocytes in the role of inflammation.   there was one reference about mycophenolate - this has potential for treatment and the article would have a larger impact if implications on treatment or at least halting the progression of disease would be relevant to cinicians reading the article.

I think that the references are excellent - but I am a clinician and would like to see more clinical references as I cited.   The authors spoke about one form of therapy , but I would like to see more.

As far as specifics about the research design, since this was an analysis of articles, I only felt the need to comment on the need to expand the literature.

Author Response

"I  would like to see an expansion on the section about therapeutics that could be used as a result of knowing about the role of monocytes in the role of inflammation.   there was one reference about mycophenolate - this has potential for treatment and the article would have a larger impact if implications on treatment or at least halting the progression of disease would be relevant to cinicians reading the article."

Thank you for bringing up this important point. Indeed, the culminating point of dissecting the importance of the monocyte in the pathology of IPF is finding druggable targets that might ultimately have clinical use. After this point was brought up by reviewer 1, we endeavored to do a deeper search of the existing literature regarding drugs and agents that have been trialed or are being tested in IPF, that specifically act on the monocyte-macrophage axis. 17 new articles were identified and have been included in the subsections "Potential therapeutic approaches" (those pertaining to IPF more generally) and at the end of "Role of monocytes in connective tissue disease ILD" (clinical trials specifically in connective tissue-associated ILD). The new text is included in red to facilitate ease of identification.

Two drugs are currently approved for IPF: pirfenidone and MMF. While they somewhat halt progression of fibrosis, they do not reverse it. Few studies have investigated how those drugs specifically act on the monocyte-macrophage axis- we included all which we could identify. This represents a gap of knowledge in the literature that deserves attention, specially since these are the only drugs routinely used to treat this disease.

"I think that the references are excellent - but I am a clinician and would like to see more clinical references as I cited.   The authors spoke about one form of therapy , but I would like to see more."

Please see above. We agree with this comment and have included new references to expand on clinical trials, and experimental drugs on pre-clinical stages that could advance to clinical studies.

"As far as specifics about the research design, since this was an analysis of articles, I only felt the need to comment on the need to expand the literature."

Thank you. We hope the above effort has enriched the content of the manuscript and made it more appealing to clinicians.

Reviewer 2 Report

Comments and Suggestions for Authors

This article reviews the role of Monocytes in the natural history of Idiophatic Pulmonary Fibrosis (IPF). The article is extensive and considers several aspects of the possible role of monocytes in the progression and prognosis of IPF. The authors review 12 articles published on the association of monocytes with IPF and other interstitial lung diseases. Later on, they discuss about the possible mechanisms involved, the cytokines that participate in the interaction between monocytes and alveolar cells, monocyte migration and differentiation and their possible role in pulmonary fibrosis progression. The interaction between monocytes and alveolar tissue could be the target of new therapeutic approaches, as also discussed in the review. Finally, the potential role of monocytes in connective tissue disease is considered. The authors conclude from the review of the published information that monocytes play a critical role in IPF natural history and that their circulating counts could be considered as a biomarker for disease progression and treatment response. Also that monocyte recruitment and reprogramming pathways represent promising therapeutic targets for the treatment of these diseases.

I consider that the review is comprehensive and well presented. It represents a valuable contribution to the IPF field because presents relevant evidence on the importance of a cell type, monocytes that have not been extensively studied in IPF progression and therapy.

I would like to mention just two points:

- Table 1 is cited by the authors at the end of the Results section but is not present in the manuscript file.

- I would suggest the inclusion of a list of abbreviations, mainly those related to the different interstitial lung diseases, to make easier to follow the review article to the non-specialized audience.

Author Response

"Table 1 is cited by the authors at the end of the Results section but is not present in the manuscript file."

The table is now included in the manuscript file. Thank you for addressing this.

"I would suggest the inclusion of a list of abbreviations, mainly those related to the different interstitial lung diseases, to make easier to follow the review article to the non-specialized audience."

We agree, and have now provided such list of abbreviations. Again thank you for comments that will facilitate ease of read of this manuscript.